# US workforce gaps in emergency management: A mixed-methods approach of demographics, capacity, and community engagement

Ananya Verma[1], Lorraine A. Schneider[2], Rita V. Burke[3]*

1 Keck School of Medicine, University of Southern California, Los Angeles, California, United States of America, 2 NYC Preparedness and Recovery Institute, ICAP, Columbia University, New York, New York, United States of America, 3 Department of Population and Public Health Sciences, Keck School of Medicine, University of Southern California, Los Angeles, California, United States of America

* rita.burke@med.usc.edu

## Abstract

### Introduction

Within emergency management, few studies have analyzed the shifting landscape of the workforce in the United States. As emergency management is an evolving field, it is important to note changes in the profession. The purpose of this study was to examine the current emergency management workforce, specifically analyzing its demographic breakdowns, organizational concerns, and community-based dynamics. The intent was to determine if the workforce is representative of the communities they work in. Disasters are known to disproportionately impact vulnerable populations, making it more important for emergency managers to be demographically and functionally diverse to effectively reach and prepare these communities.

### Methods

Using a mixed-methods approach, data were gathered through five focus group sessions, and using themes from the focus groups, a large-scale survey was designed and disseminated to seven emergency management organizations across the country. The survey collected responses for three weeks, and participants were offered the opportunity to enter a raffle for a $100 gift card. The focus group data were analyzed using Atlas.ti, and the survey data were analyzed using Microsoft Excel.

### Results

In total, 20 emergency managers participated in the focus groups, and 232 participants completed the survey. Our analysis showed high levels of concern regarding an overall lack of funding and resources within organizations. Other concerns included insufficient representation of historically underrepresented populations, limiting emergency managers' capacity to effectively engage their communities.

**Data availability statement:** All relevant data are within the manuscript and its Supporting Information files.

**Funding:** This study was funded by the Federal Emergency Management Agency (FEMA) and the author who received the grant was RVB. The organization played no role in the study design, data collection and analysis, decision to publish, or preparation of the manuscript.

**Competing interests:** The authors have declared that no competing interests exist.

## Conclusions

This study identifies key challenges within the evolving emergency management workforce, most notably limited funding, the risk of burnout, underrepresentation of minority groups in leadership, and lack of standardization. At the same time, encouraging trends are emerging, including greater gender diversity and growing participation from younger professionals. The findings provide a foundational overview to guide future research on strengthening and supporting the workforce and the communities they serve.

## Introduction

Disasters are increasing in scope and magnitude. Globally, disaster-related costs have more than doubled, increasing from USD $70–80 billion annually from 1970 to 2000 to USD $180–200 billion annually between 2001 and 2020 [1]. Similarly, the U.S. averaged $9 billion-dollar disasters (CPI-adjusted) between 1980–2024, while the annual average for 2020–2024 was $23 billion-dollar disasters (CPI-adjusted) [2]. Due to climate change, events that were once rare are becoming far more common. A person born in 2025 has an 86% chance of living through a 1-in-100-year flood, compared to just 63% for someone born in 1990 [1]. Accompanying this trend is the fact that the "impact of natural disasters is greater on disadvantaged and vulnerable populations with respect to the overall population" [3]. Those who are poorer or more socially vulnerable often bear the heaviest burdens. Researchers have emphasized that disasters tend to amplify existing social inequalities, with risk and vulnerability shaped by entrenched societal structures. Disasters frequently result from the interaction between natural hazards and underlying social, political, and economic conditions [3].

These realities underscore the urgent need for skilled, adaptive emergency management. No longer confined to logistics and response, today's emergency managers must navigate complex social, political, and economic landscapes. Their role has expanded to include risk reduction, community engagement, equity-centered planning, and long-term resilience building. In this context, emergency managers serve not just as crisis responders but as strategic leaders shaping how communities prepare for, withstand, and recover from disasters.

To date, there has yet to be a comprehensive study that examines the current landscape of the emergency management workforce across all sectors. Indeed, existing studies principally examine the capacity of state, local, tribal, and territorial (SLTT) governments, such as the 2018 county-level survey conducted by the National Association of Counties [4] and the 2025 State, Local and Territory Findings Report "Emergency Management, Organizational Structures, Staffing, and Capacity Study" by Argonne National Laboratory [5]. Historically, the profession has been dominated by a specific demographic, with older, white males comprising the majority. In a 2014 demographics survey, 80% of emergency managers were male and 94% were Caucasian [6]. Data from the U.S. Census from 2022 further corroborated this

trend, indicating over 74% of emergency management directors were white [7]. While women remain underrepresented in the field, the 2014 study showed that female emergency managers were 10 years younger than male emergency managers, while representing 26% of those under the age of 46 [6]. These findings emphasize a persistent lack of representation across gender and racial lines within emergency management. Given the essential role of emergency managers in community disaster preparedness and recovery, it is critical that the workforce reflects all demographics to better understand and address their needs. In addition, emergency management as a field has only been recently formalized. According to the 2024 "State of the Community: 5-Year Trend Report (2019-2023)," the majority of the 277 higher education emergency management programs have only been in existence for 10–15 years or less [8]. The overarching goal of this study is to understand the current composition and needs of the evolving emergency management workforce. To date, efforts to assess the emergency management workforce have been minimal, highlighting the need for additional demographic data and qualitative understanding of the challenges emergency managers face to better guide future initiatives. The findings from this study can be used to guide emergency management organizations further in their goals of supporting their communities and workforce. This includes informing capacity-building efforts, developing strategies for employee retention, and creating policies to prioritize inclusivity.

## Methods

### Study overview

The study was reviewed and received Institutional Review Board approval (UP-23–01120). In this study, a mixed method approach was applied to thoroughly evaluate the current landscape of the emergency management workforce. The first phase included focus groups within several emergency management organizations. The results of the focus groups were then used to inform development of a nationally disseminated survey.

### Qualitative data collection and analysis

A focus group guide was designed to structure the conversations, particularly highlighting areas of concern. The questions served as a facilitator, helping prompt the participants to share their experiences regarding a variety of topics. These included defining diversity, equity, and inclusion (DEI), as well as evaluating how emergency management has shifted in recent years with the formalization of the field. As emergency management can cover a variety of jobs, participants were also asked about the structure of their organizations and how their workforce has changed. The discussions concluded by addressing any challenges faced in their communities relating to emergency management. The complete list of focus group questions is included in S1 File.

### Participant selection and sampling strategies

Between January and February 2024, five focus groups were conducted, with a variation of meeting days and times to maximize the availability and diversity of participants. Invitations were distributed to members/listservs of three organizations, the International Association of Emergency Managers (IAEM), the Emergency Management Growth Initiative (EMGI), and the Bill Anderson Fund, with Eventbrite being used as a platform for registration. The selection of the focus groups was a convenience sample, so no sample size was required.

### Inclusion and exclusion criteria

Beyond their membership or affiliation within these organizations, there were no further eligibility requirements for focus group participants. Participants represented emergency management professionals working primarily in public, private, and nonprofit emergency management organizations, rather than clinicians operating in prehospital or intrahospital emergency care settings. Each session was conducted over Zoom, with a duration of 60 minutes, and had a maximum capacity of eight participants. Transcripts of the focus groups were professionally prepared using rev.com.

### Data collection and analysis

A grounded theory approach, established by Glaser and colleagues [9], was used to analyze the qualitative data. This strategy was utilized because it is a flexible approach that can manage highly exploratory scenarios where minimal information is available. To analyze the data, the discussion from each focus group was transcribed and participants were deidentified. The confidentiality of the transcripts was maintained and accessed only by the primary coder (A.V.) and principal investigator (R.V.B.). The primary coder (A.V.) established codes to summarize the data on ATLAS.ti (Version 9.1.3; ATLAS.ti Scientific Software Development GmbH, Berlin, Germany). A conventional content analysis approach was used to arrange the rich qualitative data of the focus groups. To ensure that the coding was both valid and reliable, the criteria for assigning a specific code to a section of text was systematically developed and well documented using the ATLAS.ti software. The coding scheme was refined and developed to reflect and include emerging insights throughout the coding process. To ensure continued inter-coder reliability and validity, twenty percent of the coder's cases were reviewed. Discrepancies in coding were discussed and reconciled. These codes led to the development of larger concepts. There was a simultaneous and iterative process of data collection, analysis, and theory construction that resulted in adaptive changes in the focus groups as the study progressed. The complete codebook can be found in the S2 File.

### Quantitative data collection and analysis

The survey consisted of 44 questions in total and was divided into three distinct sections. The first section gathered demographic information from participants, including questions on age, race, ethnicity, etc. The second section focused on the characteristics of emergency management organizations, including structure, resources, and DEI. The final section of the survey explored the involvement of emergency managers in their communities. The questions in the survey were created for this study based on the larger categories identified in the focus groups, while answer choices included subthemes that were identified at the highest frequency in focus group sessions, so there was no prior validation. At the conclusion of the survey, respondents had the option to provide their email address to enter a raffle for one of two $100 gift cards. The complete survey questionnaire can be found in S3 File.

### Participant selection and sampling strategies

The survey was distributed to seven organizations: The Emergency Management Growth Initiative (EMGI), the Bill Anderson Fund, the Institute for Diversity & Equity in Emergency Management (I-DIEM), the International Association of Emergency Managers (IAEM), the Federal Emergency Management Agency's (FEMA) National Emergency Management Academies alumni network, and the California Emergency Services Association (CESA).

### Inclusion and exclusion criteria

Inclusion criteria for survey eligibility was anyone currently working in the field of emergency management for at least one year. This ensured that survey respondents had an understanding of their organizations as well as experience working in EM.

### Data collection and analysis

The organizations disseminated the survey in April 2024, and it remained open to responses for three weeks. The data analysis, including frequency calculation, was conducted using Microsoft Excel.

## Results

### Focus group results

Twenty participants took part throughout the five designated focus groups (five participants in Focus Groups 1 and 2, two participants in Focus Group 3, three participants in Focus Group 4, and five participants in Focus Group 5), and

the demographic summary of the participants is presented in Table 1. Although many topics were discussed during the various sessions, there were common themes that emerged. These themes were sorted into three different overarching categories: (1) concerns about the emergency management workforce; (2) recommendations to improve the workforce in the future; and (3) definitions of diversity, equity, and inclusion and their application to emergency management.

## Concerns about the workforce

Participants in the focus groups shared various concerns they had about the landscape of the emergency management workforce. Selected quotes from the focus groups that best reflect the concerns are provided in Table 2 below. One of the most recurring concerns that was addressed by representatives of the workforce during the sessions was a lack of funding and resources in emergency management offices, leading to difficulties with employee retention and funding for community relations. Other concerns raised during the sessions included a lack of clear definitions and standards within the emergency management field, weak community relationships, language and access barriers, and differences between the public and private sectors as well as urban and rural communities.

## Recommendations for the future

Following the discussion about concerns, participants focused on potential recommendations, including creating national standards and a certification process for emergency managers, building direct connections with the community, and increasing trust with the community. Many also focused on increasing the diversity of the workforce. These recommendations included creating internship programs to get youth more involved, increasing opportunities for emergency managers

**Table 1. Demographic characteristics of focus group participants (n = 20).**

|  |  | N | Percent (%) |
|---|---|---|---|
| *Gender* | Male | 10 | 50 |
|  | Female | 9 | 45 |
|  | Other | 1 | 5 |
| *Race/Ethnicity* | Black or African American | 5 | 25 |
|  | White | 12 | 60 |
|  | Other | 4 | 20 |
| *Salary* | Less than $50,000 - $74,999 | 7 | 35 |
|  | $75,000-$124,999 | 7 | 35 |
|  | $125,000-Greater than $175,000 | 6 | 30 |
| *Years in the field* | 0-5 | 8 | 4 0 |
|  | 6-21+ | 12 | 60 |
| *Level of Education* | Some College or Associate Degree or Bachelor's Degree | 9 | 45 |
|  | Graduate, Professional Degree, or Doctorate | 11 | 55 |
| *CEM Credential* | No | 9 | 45 |
|  | Yes | 11 | 55 |
| *Sector* | Public | 13 | 65 |
|  | Other | 7 | 35 |
| *EM as First C areer* | No | 13 | 65 |
|  | Yes | 7 | 35 |
| *Region of the US* | Midwest/Northeast | 7 | 35 |
|  | South | 8 | 40 |
|  | West | 5 | 25 |

**Table 2. Selected subthemes and corresponding quotes from focus group participants.**

| Theme (subthemes) | Quote |
|---|---|
| **Concerns** | |
| Concerns: Lack of funding and resources | "I think one of the things we've heard quite a bit about over the last few years is that emergency management organizations are not resourced well. They do not have enough staffing for the demands, they don't have the right funding to be able to support the many directions that they're being sent. And in a lot of cases, they don't have the right authorities to be able to take action as demanded by their constituents." |
| Concerns: Lack of definition within EM | "We have suffered this identity crisis between, 'Are we first responders? Do we run to the scene in a car with flashy lights or are we planners? Are we facilitators?' We're a lot of things, right? But I think really understanding the needs of each of our communities, we have to start to define our field." |
| Concerns: Lack of community connections | "We do a lot of planning about and for other people. And if we don't really do a better job of including those voices and those lived experiences, then our efforts will potentially fail. Because we're basically just guessing what people need and what they want and how they want to hear from us." |
| Concerns: Lack of access to services | "I live in North Dakota and North Dakota is rural. We have less than 700,000 people in the state of North Dakota, which is the population of a small city in other states. And not only is it rural, but we're primarily elderly. And then add in the fact that it's rural, it's elderly and we have tribal nations, getting information and services to areas that don't have access to the internet or don't want to have access to the internet. They don't speak English and they don't want to speak English. So trying to figure out how we can incorporate all of that into our services is a challenge." |
| **Recommendations** | |
| Recommendations: National standards/certification | "I'd like to see a national credentialing program that is not tied to membership in any organization, that is based on meeting and achieving certain standards of performance in the field of emergency management versus training, education, and credentials. That's tied to testing." |
| Recommendations: Building community connections | "It's all about getting out of the office and into the community, getting out of your emergency management, dress and everything and getting out into the community, talking to people. And I think that's a real way of building that cooperation." |
| Recommendations: Building trust with the community | "I mean, relationships are definitely important as an emergency manager. You have to know your community for them to trust you." |
| **Definitions of DEI** | |
| Definitions of DEI: Equal opportunities | "I think that folks that have been marginalized have been marginalized because of lack of opportunity. The more we can do to open up opportunity is how we achieve a more equitable society." |
| Definitions of DEI: Provide support to those who need it | "So in response, we utilize those working groups and those hub and spoke models to identify the needs in the community and to make sure that we're responding to those needs." |
| Definitions of DEI: Reflective of the community | "I think for me, the first word that came to my mind was representative, and I'm thinking in terms of being representative of my community. So that I'm not only representing maybe different life experiences, but lived experiences. But also different points of view, different demographics, pretty much the gamut of what my community looks like should be reflected in what I'm doing or what I'm representing." |

to volunteer in the community, and adding different training opportunities for emergency managers. Selected quotes addressing these recommendations are provided in Table 2.

### Diversity, equity, and inclusion in emergency management

The final theme that emerged during the focus groups was the definition of DEI and what it means to the participants. Some of the common definitions agreed upon by many participants were providing support to the community which they represent, having a workforce that reflects the community, and most frequently mentioned, providing equal opportunities for all individuals, regardless of their identity. These quotes are shown in Table 2.

### Survey results

Two hundred fifty-eight participants responded to the survey, of which 26 were incomplete, leaving a total of 232 complete responses. Out of the separate organizations that the survey was distributed to, EMGI had a response rate of 34%, the Bill Anderson Fund had a 3% response rate, I-DIEM had 2%, IAEM had 1%, FEMA National Emergency Management Academies had 2%, and CESA had 2% as well. The overall response rate was 2%. In terms of a demographic breakdown of the participants, there was a higher percentage of male (58%) than female respondents (42%). There was a fairly even distribution of ages; however, the majority of participants were either between 35–44 (27%) or 45–54 years old (29%). An overwhelming majority of survey participants identified as white (86%), and the next most common race/ethnicity was Hispanic or Latino (9%). Over 80% of respondents reported having at least a bachelor's degree, with 55% having a professional, doctorate, or graduate degree as well. The salaries of emergency managers were also generally evenly distributed, with approximately 50% of participants earning between $75,000 and $125,000. Most of the survey participants worked in the public sector (54%). The complete set of demographic data for survey participants is provided in Table 3.

### Emergency management organizations

The next section of questions focused on evaluating and understanding the emergency management organizations, including asking about the size of the population they serve, the structure of their office, and the number of people within their office. When asked about the structure of the organizations in which they work, 53% of emergency managers reported working in urban areas, 33% reported working in suburban areas, and 14% reported working in rural areas. Participants were also asked if there was someone in their office with DEI in their job title, to which 83% said "No". Similarly, they were asked if there was someone leading DEI efforts, to which 68% said "No" (Table 4).

Participants were asked to consider various factors related to employee retention and promotion and rank them based on which they perceived as most limiting. The most limiting factor was "inadequate salary and benefits" which was ranked first by 34% of participants. The least limiting factor ranked was "poor organizational culture of the workplace environment," which was ranked eighth by 30% of participants (Table 5).

Drawing on concerns raised in the focus groups, the subsequent survey items examined the perceived impacts of funding and resource allocation. When asked how shortages of resources and funding had affected different components of their emergency management organization, participants reported that workforce expansion was most affected (39%), whereas DEI initiatives were least affected (17%). Participants were also asked to rate the overall impact of funding and resource limitations on their organization using a 1–10 scale. The most frequently selected ratings were 7 (17%) and 10 (17%), indicating that many respondents perceived funding constraints as having a substantial or extreme impact on their operations (Table 6).

The final set of questions focused on DEI within emergency management organizations. Participants were first asked whether their office incorporates DEI and, if so, in what ways. The most frequently selected practices were engaging in

**Table 3. Survey participants' demographic characteristics (n = 232).**

| | | N | Percent (%) |
|---|---|---|---|
| *Gender* | Male | 135 | 58 |
| | Female | 96 | 42 |
| | Other | 1 | 0 |
| *Age* | 18−3 4 years | 38 | 17 |
| | 35-44 years | 63 | 27 |
| | 45-54 years | 67 | 29 |
| | 55-64 years | 45 | 19 |
| | 65 and above | 19 | 8 |
| *Race/Ethnicity* | American Indian/Alaska Native | 10 | 4 |
| | Asian | 18 | 8 |
| | Black or African American | 15 | 6 |
| | Hispanic or Latino | 20 | 9 |
| | White | 199 | 86 |
| *Sexual orientation* | Bisexual | 6 | 3 |
| | Homosexual or gay | 12 | 5 |
| | Heterosexual or straight | 190 | 82 |
| | Queer | 9 | 4 |
| | Prefer not to say | 15 | 6 |
| *Disability* | Yes, I have a physical disability | 12 | 5 |
| | Yes, I have a sensory disability (e.g., visual or hearing impairment) | 9 | 4 |
| | Yes, I have a cognitive or intellectual disability | 6 | 3 |
| | Yes, I have a mental health condition | 19 | 8 |
| | Yes, I have a chronic illness or health condition | 19 | 8 |
| | No, I do not have a disability | 167 | 72 |
| *Veteran* | No | 186 | 80 |
| | Yes | 46 | 20 |
| *Level of Education* | High school degree or equivalent (e.g., GED), Some college but no degree | 13 | 6 |
| | Associate degree | 13 | 6 |
| | Bachelor's Degree | 65 | 28 |
| | Graduate degree | 128 | 55 |
| | Doctorate degree | 13 | 6 |
| *Field of Degree* | Emergency management | 105 | 45 |
| | Homeland security | 35 | 15 |
| | Public Health | 24 | 10 |
| | Public Administration | 24 | 10 |
| | Sciences | 29 | 13 |
| | Business/economics | 20 | 9 |
| | Humanities | 13 | 6 |
| | Police/Fire | 15 | 6 |
| | Other | 34 | 15 |
| *AEM/CEM* | Associate Emergency Manager (AEM)® | 10 | 4 |
| | Certified Emergency Manager (CEM)® | 74 | 32 |
| | No | 148 | 64 |
| *Current employment status* | Employed (part-time) | 12 | 5 |
| | Employed (full-time) | 209 | 90 |
| | Not employed | 11 | 5 |

*(Continued)*

| | | N | Percent (%) |
|---|---|---|---|
| Years in EM | 0-2 | 12 | 5 |
| | 3-5 | 29 | 13 |
| | 6-10 | 42 | 18 |
| | 11-15 | 51 | 22 |
| | 16-25 | 76 | 33 |
| | 26+ | 22 | 9 |
| Salary | Less than $50,000 | 17 | 7 |
| | $50,000-$74,999 | 28 | 12 |
| | $75,000-$99,999 | 55 | 24 |
| | $100,000-$124,999 | 59 | 25 |
| | $125,000-$149,999 | 40 | 17 |
| | $150,000-$175,000 | 16 | 7 |
| | Greater than $175,000 | 17 | 7 |
| Sector | Consulting | 30 | 13 |
| | Healthcare | 11 | 5 |
| | Higher education | 21 | 9 |
| | Non-profit | 13 | 6 |
| | Private sector | 18 | 8 |
| | Public sector – County | 40 | 17 |
| | Public sector – Federal | 26 | 11 |
| | Public sector – Local | 39 | 17 |
| | Public sector – State | 21 | 9 |
| | Other | 13 | 5 |
| EM First career | No | 167 | 72 |
| | Yes | 65 | 28 |
| Previous field | EMS | 14 | 8 |
| | Fire service | 21 | 13 |
| | Law enforcement | 18 | 11 |
| | Military | 22 | 13 |
| | Public health | 12 | 7 |
| | Other | 78 | 47 |
| FEMA Region | Region 1 | 11 | 5 |
| | Region 2 | 11 | 5 |
| | Region 3 | 29 | 13 |
| | Region 4 | 39 | 17 |
| | Region 5 | 23 | 10 |
| | Region 6 | 24 | 10 |
| | Region 7 | 6 | 3 |
| | Region 8 | 10 | 4 |
| | Region 9 | 56 | 24 |
| | Region 10 | 20 | 9 |
| Fluent in another language | No | 196 | 14 |
| | Yes | 33 | 86 |

*(Continued)*

**Table 3.** (Continued)

| Other language | | N | Percent (%) |
|---|---|---|---|
| *Other language* | Spanish | 20 | 9 |
| | German | 8 | 3 |
| | French | 5 | 2 |
| | Other | 12 | 5 |

**Table 4. Survey results of structure of emergency management organizations (n = 232).**

| | | N | Percent (%) |
|---|---|---|---|
| *Area of work* | Rural | 26 | 14 |
| | Suburban | 59 | 33 |
| | Urban | 96 | 53 |
| *Size of population served* | < 24,999 | 23 | 14 |
| | 25-49,999 | 27 | 10 |
| | 50,000-74,999 | 10 | 6 |
| | 75,000-149,999 | 26 | 16 |
| | 150,000-499,999 | 18 | 11 |
| | 500,000-1,000,000 | 20 | 12 |
| | > 1,000,000 | 51 | 31 |
| *Who does your EM office report to?* | Environment, Health, & Safety | 12 | 6 |
| | Fire Department | 13 | 7 |
| | Mayor/C-Suite | 41 | 21 |
| | Security | 10 | 5 |
| | Sheriff's Department | 16 | 8 |
| | Other | 101 | 52 |
| *Is the head of the EM office politically appointed?* | Yes | 45 | 23 |
| | No | 140 | 73 |
| | I don't know | 7 | 4 |
| *Number of people in EM office* | Less than 1 (it's a part-time or volunteer position) – 1 | 39 | 21 |
| | 2-5 | 72 | 38 |
| | 6-10 | 23 | 12 |
| | 11-20 | 11 | 6 |
| | 21-50 | 12 | 6 |
| | 51-100 | 9 | 5 |
| | 100+ | 25 | 13 |
| *Is there a person with DEI in their job title?* | Yes | 20 | 10 |
| | No | 159 | 83 |
| | Other | 12 | 6 |
| *Is there someone leading DEI efforts?* | Yes | 49 | 26 |
| | No | 130 | 68 |
| | Other | 12 | 6 |

**Table 5. Survey results of factors contributing to employee retention concerns (n = 232).**

| What have you found to be the most limiting factors in employee retention and promotion within your emergency management career? (rank them in order of importance – 1 being most important and 8 being least important) | | | | | | | | |
|---|---|---|---|---|---|---|---|---|
| | N (percent (%)) | | | | | | | |
| | 1 | 2 | 3 | 4 | 5 | 6 | 7 | 8 |
| Difficulty balancing work-life | 31 (16) | 31 (16) | 27 (14) | 21 (11) | 15 (8) | 27 (14) | 17 (9) | 20 (11) |
| High turnover rate among staff | 10 (5) | 16 (8) | 30 (16) | 34 (18) | 28 (15) | 20 (11) | 30 (16) | 21 (11) |
| Inadequate salary and benefits | 64 (34) | 33 (17) | 31 (16) | 17 (9) | 16 (8) | 7 (4) | 6 (3) | 15 (8) |
| Insufficient training and professional development programs | 10 (5) | 10 (5) | 10 (5) | 24 (13) | 35 (19) | 43 (23) | 34 (18) | 23 (12) |
| Lack of recognition or appreciation for contributions | 9 (5) | 23 (12) | 27 (14) | 20 (11) | 35 (19) | 38 (20) | 21 (11) | 16 (8) |
| Lack of or limited opportunities for upward mobility within an organization | 34 (18) | 47 (25) | 32 (17) | 28 (15) | 17 (9) | 14 (7) | 13 (7) | 4 (2) |
| Limited opportunities for meaningful work assignments | 6 (3) | 11 (6) | 13 (7) | 17 (9) | 31 (16) | 24 (13) | 53 (28) | 34 (18) |
| Poor organizational culture or workplace environment | 25 (13) | 18 (10) | 19 (10) | 28 (15) | 12 (6) | 16 (8) | 15 (8) | 56 (30) |

**Table 6. Survey results about impacts of emergency management funding and resource allocation (n = 232).**

| How strongly impacted is each area by a lack of resources and funding? (1 being no impact and 10 being extremely impacted) | | | | | | | | | |
|---|---|---|---|---|---|---|---|---|---|
| | N (percent (%)) | | | | | | | | |
| | 1 | 2 | 3 | 4 | 5 | 6 | 7 | 8 | 9 | 10 |
| Community preparedness efforts | 9 (5) | 11 (6) | 14 (8) | 10 (5) | 30 (16) | 25 (13) | 16 (9) | 25 (13) | 24 (13) | 22 (12) |
| DEI initiatives | 27 (17) | 16 (10) | 12 (8) | 10 (6) | 22 (14) | 13 (8) | 18 (12) | 13 (8) | 10 (6) | 14 (9) |
| Emergency response capabilities | 12 (7) | 10 (6) | 11 (6) | 16 (9) | 24 (14) | 21 (12) | 28 (16) | 25 (14) | 14 (8) | 16 (9) |
| Workforce expansion | 8 (4) | 7 (4) | 5 (3) | 4 (2) | 17 (9) | 6 (3) | 14 (8) | 25 (14) | 27 (25) | 71 (39) |

| On a scale of 1–10 (1 being no impact and 10 being extreme impact), how have funding and resource limitations impacted your emergency management organization? | | | | | | | | | |
|---|---|---|---|---|---|---|---|---|---|
| | N (percent (%)) | | | | | | | | |
| | 1 | 2 | 3 | 4 | 5 | 6 | 7 | 8 | 9 | 10 |
| | 12 (6) | 7 (4) | 9 (5) | 9 (5) | 18 (9) | 18 (9) | 32 (17) | 29 (15) | 25 (13) | 32 (17) |

community outreach and partnering with diverse organizations (61%), followed by promoting cultural competency and sensitivity among staff (57%).

When asked about their most pressing DEI concerns, respondents cited limited staff awareness or understanding of DEI issues (43%), insufficient representation of marginalized groups in decision-making (36%), and a lack of diversity in leadership roles (34%) (Table 8).

## Community engagement

The next section of questions in the survey focused on the relationship between emergency managers and their communities. When asked to rate how strongly various concerns have impacted community connections, an underrepresentation of minority groups in emergency planning was selected as the most impactful (15%). Language barriers were selected as the least impactful in terms of community relations (9%) (Table 7).

Emergency managers were also asked about DEI in terms of their community and individual beliefs. Participants were asked to select any limiting factors when taking steps towards incorporating DEI practices into their communities, and the concerns they identified included a limited awareness or understanding of DEI issues within the community (41%), as well as a lack of funding and resources dedicated to DEI initiatives (38%). Participants were asked to rank four definitions of DEI in order of importance in emergency management. The definitions were generated based on the focus group

**Table 7. Survey responses about emergency management connections to community (n = 232).**

| Rate how strongly each of the following concerns have impacted community connections: (1 being no impact and 10 being extreme impact) | | | | | | | | | | |
|---|---|---|---|---|---|---|---|---|---|---|
| | N (percent (%)) | | | | | | | | | |
| | 1 | 2 | 3 | 4 | 5 | 6 | 7 | 8 | 9 | 10 |
| *Lack of transparency with the community* | 15 (10) | 22 (15) | 16 (11) | 12 (8) | 25 (17) | 13 (9) | 12 (8) | 10 (7) | 12 (8) | 12 (8) |
| *Lack of trust with community leaders* | 15 (10) | 13 (9) | 18 (12) | 9 (6) | 15 (10) | 15 (10) | 16 (11) | 18 (12) | 17 (11) | 12 (8) |
| *Lack of trust with community members* | 13 (9) | 17 (11) | 23 (15) | 9 (6) | 24 (16) | 10 (7) | 15 (10) | 16 (11) | 9 (6) | 13 (9) |
| *Language barriers* | 16 (11) | 15 (10) | 21 (14) | 15 (10) | 23 (15) | 14 (9) | 12 (8) | 14 (9) | 9 (6) | 12 (8) |
| *Physical and technological lack of access to services* | 15 (10) | 15 (10) | 17 (11) | 22 (15) | 21 (14) | 21 (14) | 13 (9) | 12 (8) | 7 (5) | 8 (5) |
| *Socioeconomic differences* | 10 (7) | 5 (3) | 20 (13) | 15 (10) | 27 (18) | 20 (13) | 16 (11) | 15 (10) | 14 (9) | 10 (7) |
| *Underrepresentation of minority groups in emergency planning* | 13 (9) | 11 (7) | 14 (10) | 19 (13) | 11 (7) | 16 (11) | 13 (9) | 17 (12) | 11 (7) | 22 (15) |

discussions. Thirty four percent of participants ranked "equal opportunity for all voices to be heard and represented" as their most important DEI definition, and 45% responded that "reflecting the community in which you work" was the least important (Table 8).

Lastly, emergency managers were asked to rate statements about their relationship with the community on a scale from strongly disagree to strongly agree. For the statements "I feel connection to the community through my organization" and "the community is receptive to the work we do," over 60% of participants selected somewhat agree and strongly agree. The complete distribution of responses is provided in Table 9.

## Integrating qualitative and quantitative findings

Across both the focus group discussions and survey responses, several consistent patterns emerged regarding workforce challenges, organizational constraints, and the role of diversity, equity, and inclusion in emergency management. The qualitative themes helped contextualize and explain trends observed in the survey data, highlighting how workforce concerns manifest in daily practice and offering depth to the quantitative patterns.

## Workforce concerns and organizational capacity

Focus group participants repeatedly emphasized limited funding, insufficient staffing, and high turnover as major concerns. These narratives align with survey findings in which respondents identified "inadequate salary and benefits" as the most limiting factor for retention and promotion (34%) and rated funding constraints as having substantial or extreme impacts on their organization (with 17% selecting 7 and 17% selecting 10 on a 10-point scale). The qualitative accounts of resource strain—such as difficulty maintaining community relationships or expanding staff—correspond directly with the survey item identifying workforce expansion as the area most affected by funding shortages (39%). Together, these data indicate that resource limitations are not only recognized at the organizational level but also experienced acutely in day-to-day operations.

## Community engagement and representation

Qualitative discussions highlighted challenges in building trust with communities, particularly when emergency management offices lack demographic diversity or sufficient cultural competence. These themes are reflected in the survey data showing that underrepresentation of minority groups in emergency planning was perceived as the most impactful barrier to community connection (15%). Focus group comments describing language and access barriers further illuminate why survey respondents identified limited community awareness of DEI issues (41%) as a major constraint. Together, these findings suggest a reinforcing cycle in which gaps in representation hinder community engagement, which in turn affects the perceived relevance and reach of emergency management activities.

**Table 8. Survey responses about diversity initiatives within emergency management organizations (n = 232).**

| *In what ways, if any, do you see your emergency management office incorporating DEI practices? (select all that apply)* | | |
|---|---|---|
| | N | Percent (%) |
| Creating inclusive hiring practices to attract diverse candidates | 79 | 48 |
| Conducting regular diversity audits to assess progress and identify areas for improvement | 36 | 22 |
| Encouraging open dialogue and feedback on DEI initiatives within the office | 74 | 45 |
| Engaging in community outreach and partnerships with diverse organizations | 100 | 61 |
| Establishing diversity-focused affinity groups or committees | 56 | 34 |
| Implementing diversity training programs for staff | 60 | 37 |
| Incorporating DEI principles into emergency response plans and procedures | 80 | 49 |
| Promoting cultural competency and sensitivity among staff | 94 | 57 |
| Providing resources and support for underrepresented groups within the office | 38 | 23 |
| Reviewing and revising policies to ensure they promote equity and inclusion | 77 | 47 |
| *Select the most pressing concerns relating to Diversity, Equity, and Inclusion (DEI) within your organization:* | | |
| | N | Percent (%) |
| Challenges in fostering an inclusive and welcoming work environment for all employees | 29 | 18 |
| Cultural insensitivity or bias in workplace interactions and practices | 35 | 22 |
| Disparities in pay and benefits based on identity factors | 22 | 14 |
| Inadequate resources or support for DEI initiatives | 45 | 28 |
| Inequitable distribution of workload or assignments | 25 | 16 |
| Insufficient representation of marginalized groups in decision-making processes | 58 | 36 |
| Lack of diversity in leadership positions | 54 | 34 |
| Lack of support from certain individuals or groups toward DEI efforts | 33 | 21 |
| Limited awareness or understanding of DEI issues among staff | 68 | 43 |
| Unequal access to career advancement opportunities based on identity factors | 21 | 13 |
| *What, if any, have been the most limiting factors in making meaningful steps toward incorporating DEI practices in your community? (select all that apply)* | | |
| | N | Percent (%) |
| Challenges in finding common ground or consensus among diverse community groups | 18 | 13 |
| Cultural or linguistic barriers hindering effective communication about DEI topics | 32 | 24 |
| Fear of change or disruption to existing power dynamics | 45 | 33 |
| Historical prejudices or biases that impede progress | 45 | 33 |
| Inadequate representation of diverse voices in decision-making processes | 40 | 29 |
| Insufficient leadership support or commitment to DEI efforts | 39 | 29 |
| Lack of funding or resources dedicated to DEI initiatives | 51 | 38 |
| Limited awareness or understanding of DEI issues within the community | 56 | 41 |
| Resistance or pushback from community members or stakeholders | 25 | 18 |
| Structural barriers or systemic inequalities within the community | 40 | 29 |
| *Select all DEI-related trainings received:* | | |
| | N | Percent (%) |
| Bystander communication | 30 | 17 |
| Cultural competence | 97 | 54 |
| Cultivating empathy | 68 | 38 |
| Disability awareness | 92 | 51 |
| Implicit bias training | 92 | 51 |

*(Continued)*

**Table 8.** (Continued)

| In what ways, if any, do you see your emergency management office incorporating DEI practices? (select all that apply) | | |
|---|---|---|
| Inclusive hiring practices | 62 | 35 |
| Intentional inclusion | 39 | 22 |
| Microaggressions | 50 | 28 |
| Sexual harassment prevention | 152 | 85 |
| Other | 16 | 9 |

| How would you define DEI in emergency management? (rank in order of importance) | | | | |
|---|---|---|---|---|
| | N (percent (%)) | | | |
| | 1 | 2 | 3 | 4 |
| Equal opportunity for all voices to be heard and represented | 52 (34) | 28 (18) | 39 (25) | 34 (22) |
| Forming connections within the community | 46 (20) | 45 (29) | 37 (24) | 25 (16) |
| Providing support to people who need it the most | 34 (22) | 55 (36) | 39 (25) | 25 (16) |
| Reflecting the community in which you work | 21 (14) | 25 (16) | 38 (25) | 69 (45) |

**Table 9. Measuring emergency managers' relationship with communities (n = 232).**

| Select which statement best applies: | | | | |
|---|---|---|---|---|
| | I feel connected to the community through my organization | The community is aware of the work we do | The community is receptive to the work we do | I work directly with the community |
| Strongly Disagree | 11 (7%) | 23 (14%) | 3 (2%) | 14 (8%) |
| Somewhat disagree | 16 (10%) | 33 (20%) | 17 (10%) | 24 (14%) |
| Neither agree nor disagree | 37 (22%) | 30 (18%) | 35 (21%) | 30 (18%) |
| Somewhat agree | 67 (41%) | 64 (39%) | 75 (45%) | 51 (31%) |
| Strongly agree | 34 (21%) | 16 (10%) | 35 (21%) | 47 (28%) |

### DEI practices and organizational gaps

Participants in the focus groups commonly defined DEI as ensuring that emergency management offices reflect the communities they serve and provide equitable opportunities for involvement. However, many noted a lack of standardized DEI practices or leadership within their organizations. Survey data support these concerns: 83% of respondents reported having no one in their office with DEI in their job title, and 68% reported no designated person leading DEI efforts. Similarly, the most frequently selected DEI-related concerns—limited staff awareness of DEI issues (43%) and insufficient representation of marginalized groups in decision-making (36%)—mirror the qualitative narrative that DEI remains an aspirational rather than operationalized component of emergency management practice.

## Discussion

### Key findings and implications

As no prior research has examined the current emergency management workforce in the U.S. across the public, private, and nonprofit sectors, this study collected essential information from emergency managers about their organizational demographics, key concerns, and recommendations.

A prominent concern was a lack of underrepresented groups in decision-making and emergency planning processes. This finding is particularly significant in light of the survey results, which revealed that while most emergency managers believe that it is essential for all voices to be heard and represented, one of the most significant challenges to achieving that goal is the limited awareness or understanding of inclusivity issues among staff. Despite a growing emphasis on

diversity and inclusion within the field, these findings indicate a substantial gap between the goals of emergency management organizations and their current practices.

The study also captured evidence of positive shifts within the workforce. Demographic data from the surveys and focus groups indicate a gradual closing of the gender gap in emergency management leadership. Compared to a 2014 study reporting that 80% of emergency managers were male, this survey revealed a more balanced distribution, with 58% male and 42% female participants [6]. Additionally, there has been a notable shift in age demographics. Where the field was historically dominated by retirement-age professionals, our study revealed a higher representation of younger emergency managers, with most participants under 55. However, racial diversity remains a critical concern, as an overwhelming majority of emergency managers identified as White. In a further stratified analysis to understand how intersectionality could shape emergency managers' experiences in the field, we found that the largest group of survey participants identified as white males, which is also consistent with the current demographic breakdown of the emergency management workforce [6].

While our results highlight the importance emergency managers place in community engagement, there is limited existing literature within the emergency management field on how community representation and engagement can improve outcomes. In contrast, other disciplines, including health research ethics and medical care, show community engagement can improve the work they are doing. In the field of mental health research, a study conducted by Dubois et al. showed that community-engaged research can enhance study quality, strengthen partnerships, and address ethical concerns by involving community advisory boards and community members as co-investigators [10]. Integrating opinions of community members into research work not only improved the quality of the research conducted but increased the engagement of the community with the research outcomes and their knowledge of the topics. In healthcare, research demonstrates that the absence of community representation within medical providers can lead to inequitable and harmful policies. Ramsey and Najarian (2024) argue that meaningful representation of marginalized groups directly improves decision-making and service delivery, emphasizing that systems without representation risk perpetuating discriminatory practices [11]. By increasing representation, specifically of underrepresented groups of the community, organizations can more effectively reach those in most need of assistance.

Together, our findings and the broader interdisciplinary literature suggest that greater integration of community engagement frameworks into emergency management research and practice represents a critical opportunity to advance the field. Future research can continue to build on this gap by examining engagement practices in emergency management and using established knowledge from related disciplines.

## Strengths and limitations

This study has several limitations. The survey response rate was low, suggesting that the results may not be generalizable to other populations. In addition, while the survey was distributed to many organizations, the responses were not evenly distributed and most of the participation came from two specific organizations, which could further cause bias. There are many factors that could have contributed to the lower response rate, including survey fatigue, competing responsibilities, and limitations in our dissemination method. Furthermore, the survey questions were created specifically for this research study and were not pilot tested prior to dissemination, which could lead to bias in the responses as well.

Despite the limitations, this study has several strengths. The mixed-methods approach combined the insights from the focus groups with the quantitative data from the nationally disseminated survey. Additionally, it is the only study to date, to our knowledge, that examines the emergency management workforce across all sectors (government (SLTT), private, and nonprofit) and geographic distinctions (urban, suburban, and rural).

## Recommendations for the future

There are many future directions for research in this area. First, future studies can be conducted to understand more of the specific impacts of funding limitations, delving into their impact on workforce retention, job satisfaction, and community

relationships. Further research can also be conducted on the standardization of the industry, which was a common concern mentioned in the focus group sessions. Building on Patel et al.'s study [12] another area of future research could focus on the mental health of emergency managers, including burnout, particularly with the increasing complexity of disasters, as well as to develop strategies and practices to promote well-being and resilience. Furthermore, the standardization of emergency management degree programs through the use of core competencies, like sociocultural literacy [13], can equip future emergency managers to better engage with diverse populations. Lastly, emergency management agencies could follow the approach FEMA originally pursued in its 2022–2026 Strategic Plan [14] to prioritize and harness a workforce that increasingly reflects the community they serve.

Focus group participants proposed strategies to strengthen community ties, enhance workforce diversity, and professionalize the field, such as national standards, certification processes, mentorship opportunities, and youth engagement programs. While the survey did not directly measure these recommendations, the strong agreement among respondents that their community is receptive to the work they do (over 60% somewhat or strongly agreed) suggests potential readiness for implementing community-centered or partnership-based approaches. Moreover, the low frequency of respondents identifying DEI initiatives as heavily impacted by funding constraints (17%) may indicate organizational willingness to maintain DEI-related work despite broader resource challenges.

## Conclusion

The emergency management workforce is constantly changing and developing, and with that comes a need to understand areas of improvement and concern. This study highlights current critical challenges facing the workforce, including limitations on funding, which may lead to employee burnout, underrepresentation of minority groups in decision-making, and a lack of standardization. There are, however, also many signs of improvement and development which can be seen in an increased gender diversity and an increase in younger individuals. This study represents an initial step, laying the groundwork for future research aimed at further elucidating and addressing these specific challenges.

## Supporting information

**S1 File. Focus Group Guide.**
(PDF)

**S2 File. Focus Group Codebook.**
(PDF)

**S3 File. Survey Questions.**
(DOCX)

**S1 Data. Deidentified dataset.**
(XLSX)

## Author contributions

**Conceptualization:** Rita V. Burke.

**Data curation:** Ananya Verma.

**Formal analysis:** Ananya Verma.

**Funding acquisition:** Rita V. Burke.

**Project administration:** Rita V. Burke.

**Supervision:** Lorraine A. Schneider, Rita V. Burke.

**Writing – original draft:** Ananya Verma.

**Writing – review & editing:** Lorraine A. Schneider, Rita V. Burke.

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
