## [Decision Letter · Decision Letter 0]

12 Jun 2025

Dear Dr. Burke,

Thank you for submitting your manuscript to PLOS ONE. After careful consideration, we feel that it has merit but does not fully meet PLOS ONE’s publication criteria as it currently stands. Therefore, we invite you to submit a revised version of the manuscript that addresses the points raised during the review process.

**ACADEMIC EDITOR:**

We look forward to receiving your revised manuscript.

Kind regards,

Javier Fagundo-Rivera, PhD

Academic Editor

PLOS ONE

**Additional Editor Comments:**

Dear Authors,

The article addresses a gap in the research on the current landscape of the emergency management workforce, which, according to the authors, has not been the subject of a comprehensive recent study.

The mixed-methods approach allows for a deeper exploration than prior studies that focused solely on demographic aspects.

The article is generally well organized, following standard sections.

The abstract and introduction clearly reflect the study's objective (examining the emergency management workforce, including demographics, organizational concerns, and community relations) and its significance (an evolving field, the need for a workforce that reflects the communities it serves, and the lack of prior research).

General comments:

1) Information is provided regarding the topics covered in the focus groups and the survey sections. However, the exact wording of the questions used in both instruments is not fully available in the provided materials, which may limit the precise replication of the study. To improve replicability, the authors are encouraged to consider making the full questionnaires for both the focus groups and the survey available, possibly as supplementary supporting material.

These changes are substantial as they directly affect the validity of the conclusions drawn from the survey data and the transparency of the analytical methods employed, in line with PLOS ONE standards. Once these issues are addressed, the manuscript will have a stronger foundation for publication.

2) The methodology section mentions the use of logistic regression to identify significant relationships between variables; however, the results of this analysis are not presented anywhere in the Results section or in the tables. This constitutes a notable inconsistency between what is reported in the Methods and what is shown in the Results. The authors should either include the results of the logistic regression or, alternatively, remove the reference to this analysis from the Methods section.

3) The authors state that a "nationally representative" survey was conducted. However, in the Discussion section, they mention a low response rate (2%) and that the majority of responses came from only two organizations. These factors significantly limit the generalizability of the survey results and call into question the claim of national representativeness as currently stated. The authors should revise the description of the survey to accurately reflect these limitations, potentially qualifying its representativeness. In addition, the discussion on the implications of this limitation for the generalizability of the survey findings should be expanded and approached with greater caution.

Results:

The results from the focus groups (common themes, selected quotations) and the survey (descriptive statistics presented in tables) are clearly presented and consistent with the descriptive methods employed (thematic analysis for focus groups, frequencies/percentages for the survey). As previously mentioned, while the Methods section describes the use of statistical analyses such as logistic regression, the results of such analyses are not presented in the provided materials.

Recommendation:

Based on the evaluation, the article presents valuable information and addresses an important topic using a suitable mixed-methods approach. However, there are significant issues regarding the presentation of the methodology and results, as well as in the interpretation of the survey’s representativeness, which affect the quality and validity of certain key claims.

**Reviewers' comments:**

Reviewer's Responses to Questions

**Comments to the Author**

1. Is the manuscript technically sound, and do the data support the conclusions?

Reviewer #1: No

Reviewer #2: Partly

2. Has the statistical analysis been performed appropriately and rigorously?

Reviewer #1: No

Reviewer #2: Yes

3. Have the authors made all data underlying the findings in their manuscript fully available?

Reviewer #1: Yes

Reviewer #2: No

4. Is the manuscript presented in an intelligible fashion and written in standard English?

Reviewer #1: Yes

Reviewer #2: Yes

**Reviewer #1:**

Dear authors,

Given the importance of the subject of disasters, the following points can be useful in improving your valuable article.

The abstract should be properly based on the structure of the journal, including introduction, method, results, and findings.

Keywords should be selected based on the network.

The method should be stated more precisely in the abstract.

The purpose of the research should be properly stated in the abstract and introduction.

In the introduction:

The topic is not well-explained. The scope of the increase in disasters and their consequences, based on international reports, is not mentioned. Global and American statistics should be referred to, and then the situation of the United States should be examined.

The research gap in the present study is not well explained.

What are the findings of your study?

The method is not well-explained. How to select people. Criteria and input, and output flow. What is the sample size formula? Questions related to this topic have not been answered. It is suggested that each of the questions in this section about the focus group method be presented, and it is suggested that abbreviations be used in the text. The sample size in this type of study should be mentioned with the source. Why is it this number? The article lacks a source in the method. It should be corrected.

This article does not include a discussion on the topic of the article. Conclusions based on content are not considered a discussion.

**Reviewer #2:**

This article presents an original and timely investigation into the demographics, structural challenges, and community engagement dynamics of the current emergency management (EM) workforce in the United States. The study addresses an underexplored but increasingly relevant aspect of public health and disaster preparedness using a mixed-methods approach that involves both focus groups and a national survey. Therefore, I provide my evaluation following journal criteria.

**Criterion 1. The study presents the results of original research**

The authors report the findings of both qualitative focus groups and a nationally distributed survey, developed and implemented as part of this project. The integration of qualitative insights with quantitative data enhances the originality of the study and adds depth to its conclusions. The work represents a novel and valuable contribution to the emergency management and public health literature.

**Criterion 2. Results reported have not been published elsewhere**

There is no indication that the results have been previously published. The data are recent (2024), and the scope and synthesis of the mixed-method approach appear unique.

**Criterion 3. Experiments, statistics, and analyses are performed to a high standard and are described in sufficient detail**

The general methodology is appropriate for the research objectives. However, several areas lack sufficient detail or technical depth. While Atlas.ti was used for coding, the manuscript does not describe the coding process, development of the codebook, inter-coder reliability, or whether the thematic analysis followed a specific framework. The manuscript does not state whether the survey was tested or validated before deployment. Some questions may be susceptible to bias without that validation step. Frequency and logistic regression are briefly mentioned, but no regression results (e.g. odds ratios, confidence intervals, p-values) are reported. This significantly limits interpretability and undermines the credibility of quantitative inferences.

Expand the Methods section to detail the survey design process (including item construction and pilot testing). Report logistic regression outputs or remove the claim of inferential analysis. Clarify how qualitative themes were derived and supported with rigor.

**4. Conclusions are supported by the data and presented appropriately**

The conclusions are well-aligned with the data. The manuscript effectively integrates focus group narratives with survey responses to support broader claims about funding challenges, workforce composition, and DEI efforts. In particular, the discussion is balanced in acknowledging strengths (e.g., improving gender representation) and weaknesses (e.g., persistent racial homogeneity) within the EM workforce.

However, some findings, such as improving gender diversity, could benefit from more nuance, especially given the modest sample sizes and potential non-response bias.

**5. The article is intelligible and written in standard English**

The manuscript is mostly well-written and accessible. Nevertheless, several minor grammatical errors and repetitive sentence structures (e.g., "age" repeated twice in one sentence; inconsistent verb tenses) detract slightly from readability. A thorough copyediting pass is recommended to improve sentence flow and technical precision.

**6. The research meets ethical standards and is ethically sound**

The study received IRB approval from the University of Southern California (#UP-23-01120). The authors appropriately used information sheets for informed consent and followed exempt research protocols for minimal-risk studies. No ethical concerns are apparent.

**7. The article adheres to reporting guidelines and community data availability standards**

The data availability statement affirms that all relevant data are available in the manuscript and supporting materials. This is consistent with the PLOS ONE policies. Ethical disclosures and funding disclosures are complete and transparent.

**Additional Comments and Recommendations**

Low Response Rate (2%): The manuscript should provide a stronger justification for this low survey response rate and discuss the potential for selection bias. For example, overrepresentation from a few organizations may skew results toward specific perspectives.

Sample Representativeness: The study claims to use a 'nationally representative' survey, but does not describe how representativeness was ensured. Consider adjusting this claim or providing a clearer sampling strategy and weighting information.

Clarity on Inclusion/Exclusion Criteria: Neither the qualitative nor quantitative sections provide details on participant eligibility criteria beyond membership in EM organizations. This should be explicitly stated.

Consideration of Intersectionality: While demographic breakdowns are extensive, analysis could be enriched by addressing intersectional experiences, for instance, how gender and race/ethnicity together shape experiences in EM.

Future Directions Section: This is well developed and proposes compelling avenues for further work, such as mental health in EM and standardization of credentials. The authors may also consider suggesting policy implications from their findings.

The manuscript offers important contributions to the literature on workforce development, equity, and operational challenges in EM. Although the study is well-conceived and its conclusions are largely supported by the data, several methodological and reporting issues must be addressed before the manuscript can be recommended for publication.

**Do you want your identity to be public for this peer review?** For information about this choice, including consent withdrawal, please see our Privacy Policy

Reviewer #1: No

Reviewer #2: No

---

## [Author Response · Author response to Decision Letter 1]

2 Oct 2025

Response to Reviewers

Author Response:

We thank the editor and reviewers for their rigorous feedback. We would like to also thank the editor for the opportunity to revise and resubmit the article. We believe that the changes that have been made throughout have made for a stronger submission.

Journal Requirements:

Requirement 1: Please ensure that your manuscript meets PLOS ONE's style requirements, including those for file naming. The PLOS ONE style templates can be found at

Response 1: Thank you for the reminder. We have reviewed our paper to ensure that it meets the journal’s style requirements.

Requirement 2: We note that your Data Availability Statement is currently as follows: [All relevant data are within the manuscript and its Supporting Information files.]

Response 2: Thank you for this comment. We have included all of the aggregated data in our analysis. For some of the responses, the numbers are very low (<5) that risk the identification of the individuals who participated and would compromise their confidentiality. In addition, our institutional review board approval did not include make the dataset available.

Additional Editor Comments:

Comment: Information is provided regarding the topics covered in the focus groups and the survey sections. However, the exact wording of the questions used in both instruments is not fully available in the provided materials, which may limit the precise replication of the study. To improve replicability , the authors are encouraged to consider making the full questionnaires for both the focus groups and the survey available, possibly as supplementary supporting material.

These changes are substantial as they directly affect the validity of the conclusions drawn from the survey data and the transparency of the analytical methods employed, in line with PLOS ONE standards. Once these issues are addressed, the manuscript will have a stronger foundation for publication.

Response: Thank you for your feedback.The full focus group and survey questionnaires have been added as supplementary supporting material.

Comment 2: The methodology section mentions the use of logistic regression to identify significant relationships between variables; however, the results of this analysis are not presented anywhere in the Results section or in the tables. This constitutes a notable inconsistency between what is reported in the Methods and what is shown in the Results. The authors should either include the results of the logistic regression or, alternatively, remove the reference to this analysis from the Methods section.

Response: Thank you for your feedback. The reference to this analysis from the Methods section has been removed.

Comment: The authors state that a "nationally representative" survey was conducted. However, in the Discussion section, they mention a low response rate (2%) and that the majority of responses came from only two organizations. These factors significantly limit the generalizability of the survey results and call into question the claim of national representativeness as currently stated. The authors should revise the description of the survey to accurately reflect these limitations, potentially qualifying its representativeness. In addition, the discussion on the implications of this limitation for the generalizability of the survey findings should be expanded and approached with greater caution.

Response: Thank you for this comment. The description of the survey was altered from ‘nationally representative survey’ to ‘nationally disseminated survey’ as we believe that more accurately represents the applicability of the survey. As mentioned by the reviewers, the low response rate may compromise the generizability of the survey results, so we also additionally expanded on that in the limitations section in the discussion.

Comment: Results: The results from the focus groups (common themes, selected quotations) and the survey (descriptive statistics presented in tables) are clearly presented and consistent with the descriptive methods employed (thematic analysis for focus groups, frequencies/percentages for the survey). As previously mentioned, while the Methods section describes the use of statistical analyses such as logistic regression, the results of such analyses are not presented in the provided materials.

Recommendation: Based on the evaluation, the article presents valuable information and addresses an important topic using a suitable mixed-methods approach. However, there are significant issues regarding the presentation of the methodology and results, as well as in the interpretation of the survey’s representativeness, which affect the quality and validity of certain key claims.

Reviewers' Comments:

Reviewer #1:

Dear authors,

Given the importance of the subject of disasters, the following points can be useful in improving your valuable article.

Comment: The abstract should be properly based on the structure of the journal, including introduction, method, results, and findings.

Response: Thank you for your feedback. The abstract was separated into different sections: Introduction, Method, and Results/Findings. It follows all listed requirements by the journal.

Comment: Keywords should be selected based on the network.

Response: Thank you for your comment. The following keywords have been added: “emergency management, workforce development, capacity building.”

Comment: The method should be stated more precisely in the abstract.

Response: Thank you for your feedback. The abstract was revised to include a broader explanation of the method. We incorporated more detail into who the survey was disseminated to, the period of time it was open for, and the incentivization used for the survey.

Comment: The purpose of the research should be properly stated in the abstract and introduction.

Response: Thank you for this comment. We expanded on the purpose in the abstract and in the introduction.

Comment: In the introduction: The topic is not well-explained. The scope of the increase in disasters and their consequences, based on international reports, is not mentioned. Global and American statistics should be referred to, and then the situation of the United States should be examined.

Response: Thank you for this comment. We have now included the scope of the increase in disasters and their consequences using both US and international statistics and references.

Comment: The research gap in the present study is not well explained.

Response: Thank you for this comment. We have added a sentence in the Introduction to better frame the lack of research in this area and the need for our submitted study.

Comment: What are the findings of your study?

The method is not well-explained. How to select people. Criteria and input, and output flow. What is the sample size formula? Questions related to this topic have not been answered. It is suggested that each of the questions in this section about the focus group method be presented, and it is suggested that abbreviations be used in the text. The sample size in this type of study should be mentioned with the source. Why is it this number? The article lacks a source in the method. It should be corrected.

Response: Thank you for this comment. We have clarified how the selection process for the focus group occurred in the Method section, and explained our method further..

Comment: This article does not include a discussion on the topic of the article. Conclusions based on content are not considered a discussion.

Response: Thank you for this important observation. We had included this point in the Introduction and reiterated it again in the Discussion section.

Reviewer #2:

Comment: While Atlas.ti was used for coding, the manuscript does not describe the coding process, development of the codebook, inter-coder reliability, or whether the thematic analysis followed a specific framework.

Response: Thank you for pointing this out. We have added details regarding the coding process in the Methods section.

Comment: The manuscript does not state whether the survey was tested or validated before deployment. Some questions may be susceptible to bias without that validation step.

Response 15: Thank you for this comment. We added information in the Discussion in the limitations section expanding on this point.

Comment: Frequency and logistic regression are briefly mentioned, but no regression results (e.g. odds ratios, confidence intervals, p-values) are reported. This significantly limits interpretability and undermines the credibility of quantitative inferences.

Response: Thank you for this comment. We have removed any reference to regression.

Comment: Expand the Methods section to detail the survey design process (including item construction and pilot testing).

Response Thank you for this comment. We have added additional details regarding the survey design process in the Methods section.

Comment: Report logistic regression outputs or remove the claim of inferential analysis.

Response: Thank you for this comment. We have removed any reference to regression analysis.

Comment: Clarify how qualitative themes were derived and supported with rigor.

Response: Thank you for this point. We have added clarification about the qualitative analysis in the Methods section.

Conclusions are supported by the data and presented appropriately

The conclusions are well-aligned with the data. The manuscript effectively integrates focus group narratives with survey responses to support broader claims about funding challenges, workforce composition, and DEI efforts. In particular, the discussion is balanced in acknowledging strengths (e.g., improving gender representation) and weaknesses (e.g., persistent racial homogeneity) within the EM workforce.

Comment: However, some findings, such as improving gender diversity, could benefit from more nuance, especially given the modest sample sizes and potential non-response bias.

Response: Thank you for this thoughtful feedback. We agree and have revised the Discussion to add nuance around the findings on gender diversity, noting the modest sample size and potential non-response bias. We now frame these findings as preliminary observations rather than definitive trends.

The article is intelligible and written in standard English

The manuscript is mostly well-written and accessible. Nevertheless, several minor grammatical errors and repetitive sentence structures (e.g., "age" repeated twice in one sentence; inconsistent verb tenses) detract slightly from readability.

Comment: A thorough copyediting pass is recommended to improve sentence flow and technical precision.

Response: Thank you for this comment. We have gone through the manuscript to refine the flow and technical precision.

Additional Comments and Recommendations

Comment: Low Response Rate (2%): The manuscript should provide a stronger justification for this low survey response rate and discuss the potential for selection bias. For example, overrepresentation from a few organizations may skew results toward specific perspectives.

Response: Thank you for raising this important point. We acknowledge that the survey yielded a low response rate (2%), which limits the generalizability of the findings. This is a recognized challenge in surveys targeting emergency management professionals, who often have demanding schedules and limited availability for voluntary research participation.

Several factors may have contributed to the low response, including competing responsibilities, survey fatigue, and limitations in the dissemination method. While we cannot eliminate the possibility of response bias, for example, that those more engaged with or interested in the topic may have been more likely to respond, we believe the responses provide valuable insight into the current state of emergency management.

We have expanded the limitations section to reflect these concerns and to clarify that the findings are best viewed as exploratory and hypothesis-generating. Despite the low response rate, the data offer a starting point for future research and help to illuminate areas for further inquiry and policy development.

Comment: Sample Representativeness: The study claims to use a 'nationally representative' survey, but does not describe how representativeness was ensured. Consider adjusting this claim or providing a clearer sampling strategy and weighting information.

Response: Thank you for this comment. We have revised the claim of ‘nationally representative’ to more accurately describe the survey.

Comment: Clarity on Inclusion/Exclusion Criteria: Neither the qualitative nor quantitative sections provide details on participant eligibility criteria beyond membership in EM organizations. This should be explicitly stated.

Response: Thank you for this comment. We have added additional details regarding eligibility criteria in the Methods section.

Comment: Consideration of Intersectionality: While demographic breakdowns are extensive, analysis could be enriched by addressing intersectional experiences, for instance, how gender and race/ethnicity together shape experiences in EM.

Response: Thank you for this comment. We have conducted additional stratified analyses and briefly discuss the main findings in the Discussion section.

Comment: Future Directions Section: This is well developed and proposes compelling avenues for further work, such as mental health in EM and standardization of credentials. The authors may also consider suggesting policy implications from their findings.

Response: Thank you for this comment. We have added a short section in the Discussion section to address this important point.

Comment: The manuscript offers important contributions to the literature on workforce development, equity, and operational challenges in EM. Although the study is well-conceived and its conclusions are largely supported by the data, several methodological and reporting issues must be addressed before the manuscript can be recommended for publication.

Response: Thank you for this point. We have addressed the methodological and reporting issues that the reviewers have pointed out throughout the manu

---

## [Decision Letter · Decision Letter 1]

14 Oct 2025

Dear Dr. Burke,

Thank you for submitting your manuscript to PLOS ONE. After careful consideration, we feel that it has merit but does not fully meet PLOS ONE’s publication criteria as it currently stands. Therefore, we invite you to submit a revised version of the manuscript that addresses the points raised during the review process.

Thank you for submitting the revised version of your manuscript, which has now been re-evaluated by the reviewers who assessed your initial submission.

After careful consideration of their reports, it is clear that the major concerns raised in the previous review round have not been adequately addressed. The reviewers note that key issues regarding the clarity and focus of the introduction, the methodological classification, and the depth and coherence of the discussion remain largely unresolved.

Although some revisions were made, they are considered mostly superficial and do not substantially improve the manuscript’s analytical rigor or methodological transparency. In particular, the reviewers emphasize that the integration between the qualitative and quantitative components remains weak, and that the qualitative analysis lacks sufficient explanation and depth. Furthermore, the current version still does not provide a clear or well-supported conclusion in the abstract.

Given these persistent concerns, the reviewers recommend that the manuscript requires major revision before it can be considered for publication. We therefore invite you to carefully address each of the reviewers’ comments in detail, providing substantial methodological clarification, improving the structure and focus of your manuscript, and enhancing the interpretive depth of your discussion.

Should you decide to submit a further revision, please ensure that all changes are clearly indicated and justified in your response letter.

Thank you again for your interest in publishing with PLOS ONE. We appreciate your continued effort to improve your manuscript.

Kind regards.

We look forward to receiving your revised manuscript.

Kind regards,

Javier Fagundo-Rivera, PhD

Academic Editor

PLOS ONE

**Journal Requirements:**

**Additional Editor Comments:**

Dear Authors,

Thank you for submitting the revised version of your manuscript, which has now been re-evaluated by the reviewers who assessed your initial submission.

After careful consideration of their reports, it is clear that the major concerns raised in the previous review round have not been adequately addressed. The reviewers note that key issues regarding the clarity and focus of the introduction, the methodological classification, and the depth and coherence of the discussion remain largely unresolved.

Although some revisions were made, they are considered mostly superficial and do not substantially improve the manuscript’s analytical rigor or methodological transparency. In particular, the reviewers emphasize that the integration between the qualitative and quantitative components remains weak, and that the qualitative analysis lacks sufficient explanation and depth. Furthermore, the current version still does not provide a clear or well-supported conclusion in the abstract.

Given these persistent concerns, the reviewers recommend that the manuscript requires major revision before it can be considered for publication. We therefore invite you to carefully address each of the reviewers’ comments in detail, providing substantial methodological clarification, improving the structure and focus of your manuscript, and enhancing the interpretive depth of your discussion.

Should you decide to submit a further revision, please ensure that all changes are clearly indicated and justified in your response letter.

Thank you again for your interest in publishing with PLOS ONE. We appreciate your continued effort to improve your manuscript.

Kind regards.

Reviewers' comments:

Reviewer's Responses to Questions

**Comments to the Author**

Reviewer #1: (No Response)

Reviewer #2: All comments have been addressed

2. Is the manuscript technically sound, and do the data support the conclusions?

Reviewer #1: No

Reviewer #2: Partly

3. Has the statistical analysis been performed appropriately and rigorously?

Reviewer #1: No

Reviewer #2: Yes

4. Have the authors made all data underlying the findings in their manuscript fully available?

Reviewer #1: Yes

Reviewer #2: No

5. Is the manuscript presented in an intelligible fashion and written in standard English?

Reviewer #1: Yes

Reviewer #2: Yes

**Reviewer #1:**

Dear Author,

Thank you for your efforts, please review the following again.

Abstract: There is no conclusion in this study.

In the keywords, surge capacity is a more accurate option in responding to crises.

In the introduction, the willingness of the authorities to provide in 2024 and 2025 is presented. In this study, the role of gender is discussed more. If this study is about the role of human resources and the role of its management in crises, this introduction should focus on the pre-hospital topic. This can also affect the title of this study.

In the introduction, the focus of the topic is not well explained.

Studies conducted show that there is no precise classification in the method. This article can provide a better presentation in a classification method.

The discussion in this study is not well explained, and the changes are not made clearly

**Reviewer #2:** SEE DOCUMENT ATTACHED

The manuscript explores the composition and challenges of the emergency management workforce using a mixed-methods design. The topic is relevant and timely, and the study aims to fill a clear gap in understanding workforce diversity and organizational issues within this field. However, while the paper presents descriptive results with clarity, the methodological integration between qualitative and quantitative components is weak, and the qualitative analysis lacks depth and transparency. The manuscript would benefit from a more rigorous approach to mixed-methods reporting, as well as greater methodological detail and interpretive caution in its conclusions. Therefore, I recommend major revision before the manuscript can be considered suitable for publication.

**Do you want your identity to be public for this peer review?** For information about this choice, including consent withdrawal, please see our Privacy Policy

Reviewer #1: No

Reviewer #2: No

---

## [Author Response · Author response to Decision Letter 2]

28 Nov 2025

Response to Reviewers

Comments to the Author

2. Is the manuscript technically sound, and do the data support the conclusions?

Reviewer #1: No

Reviewer #2: Partly

3. Has the statistical analysis been performed appropriately and rigorously?

Reviewer #1: No

Reviewer #2: Yes

4. Have the authors made all data underlying the findings in their manuscript fully available?

Reviewer #1: Yes

Reviewer #2: No

5. Is the manuscript presented in an intelligible fashion and written in standard English?

Reviewer #1: Yes

Reviewer #2: Yes

6. Review Comments to the Author

Reviewer #1:

Dear Author,

Thank you for your efforts, please review the following again.

Comment: Abstract: There is no conclusion in this study.

Response: Thank you for this comment. We have added a concluding paragraph at the end of the abstract.

Comment: In the keywords, surge capacity is a more accurate option in responding to crises.

Response: Thank you for this comment. We have added “surge capacity” to the key words.

Comment: In the introduction, the willingness of the authorities to provide in 2024 and 2025 is presented. In this study, the role of gender is discussed more. If this study is about the role of human resources and the role of its management in crises, this introduction should focus on the pre-hospital topic. This can also affect the title of this study.

Response: Thank you for this comment. We have revised the introduction to ensure that the focus of the study is clear.

Comment: In the introduction, the focus of the topic is not well explained.

Response: Thank you for bringing this to our attention. We have added several sentences to better focus the introduction.

Comment: Studies conducted show that there is no precise classification in the method. This article can provide a better presentation in a classification method.

Response: Thank you for this comment.

Comment: The discussion in this study is not well explained, and the changes are not made clearly.

Response: Thank you. We have revised and edited the discussion for clarity.

Reviewer #2: SEE DOCUMENT ATTACHED

The manuscript explores the composition and challenges of the emergency management workforce using a mixed-methods design. The topic is relevant and timely, and the study aims to fill a clear gap in understanding workforce diversity and organizational issues within this field.

Comment: However, while the paper presents descriptive results with clarity, the methodological integration between qualitative and quantitative components is weak, and the qualitative analysis lacks depth and transparency. The manuscript would benefit from a more rigorous approach to mixed-methods reporting, as well as greater methodological detail and interpretive caution in its conclusions. Therefore, I recommend major revision before the manuscript can be considered suitable for publication.

Response: Thank you for bringing this to our attention. We have added detail to the Methods section to better explain how the focus groups informed the development of the survey. We also reanalyzed our qualitative transcripts to add depth to the results.

Originality and Relevance:

The study presents original research not previously published elsewhere, and its focus on workforce composition, diversity, and retention in emergency management contributes to an underexplored area.

Comment: However, the originality is somewhat limited by the descriptive nature of both data sets and by the absence of strong analytical connections between them. The paper would gain impact if it moved beyond summary reporting to interpretive synthesis, linking qualitative findings to survey patterns or using focus group insights to explain observed trends.

Response: Thank you for this helpful comment. We have added a section in the Results titled "Integrating Qualitative and Quantitative Findings” to facilitate the synthesis of the two types of results.

Technical and Analytical Standards:

The quantitative part of the study is straightforward and replicable. Descriptive statistics are adequate and the tables are clear.

Comment: The paper should acknowledge that the quantitative findings remain exploratory and descriptive rather than inferential.

Response: Thank you. We have added a sentence in the Discussion that acknowledges this point.

Comment: The qualitative component raises more serious concerns. The methods section states that Atlas.ti was used for coding, but it does not describe how the codebook was developed, whether coding was iterative, or whether inter-coder reliability was established. The absence of references to analytical frameworks (for example, Braun and Clarke’s thematic analysis or grounded theory procedures) weakens the credibility of the thematic synthesis. Selected quotes are presented, but they serve as illustrations rather than evidence of systematic interpretation. The analysis reads as a set of themes compiled from responses rather than the outcome of rigorous qualitative inquiry.

Response: Thank you for this important comment. We agree that additional details are necessary to describe the qualitative methods and analysis and have provided them in the methods section. We have also revised the Results section to reflect a more rigorous analytical approach.

Comment: Given that this is a mixed-methods study, a clear explanation of how qualitative and quantitative data informed each other is missing. The paper describes the focus groups as informing the survey design, but there is no evidence of integration in the analysis or discussion phases. To meet mixed-methods standards, the authors should explicitly describe the rationale for combining methods, the point of integration, and how qualitative insights complement or explain survey findings. For a better understanding of the required integration of quantitative and qualitative data, I would suggest reading this article: https://www.sciencedirect.com/science/article/pii/S0959475225001367

Response: Thank you for this point and for the article. We have added a more detailed explanation to the methods section to explain how the docs group helped inform the development of the survey.

Conclusions and Support by Data:

The conclusions are generally consistent with the data but sometimes overstate the strength and generalizability of the results.

Comment: The claim that the study describes the “current landscape” of emergency management in the United States is overstated given the 2 % survey response rate and the concentration of responses within two organizations. The discussion appropriately notes this limitation, yet some interpretive statements (for example, regarding progress in gender diversity) still read as definitive rather than tentative. The authors should reframe these claims as preliminary observations drawn from a small and potentially biased sample.

Response: Thank you for this. We have adjusted the title accordingly.

Clarity and Writing Quality:

Comment: The manuscript is intelligible and written in standard English, though the prose would benefit from professional copyediting to correct minor redundancies and tense inconsistencies.

Response: Thank you for this comment. We have reviewed the prose and conducted an intensive copyediting to ensure consistency and removal of any redundancy.

Comment: The organization of sections is logical, and tables are well formatted. However, transitions between qualitative and quantitative results could be smoother, emphasizing how each contributes to the overall argument.

Response: Thank you for this comment. We have added transitions between qualitative and quantitative data and added a section that discusses the synergy between the two sets of data.

Ethical and Reporting Standards:

The study received IRB approval and follows ethical standards for research involving human participants. The authors acknowledge restrictions on data sharing due to confidentiality risks, which is acceptable if justified to the journal.

Comment: However, PLOS ONE requires that minimal data sufficient for replication be made available. The authors should verify that aggregated or anonymized data supporting each table are included as supplementary material.

Response: Thank you for the reminder. We have submitted an IRB amendment to allow for the sharing of aggregated data and will be happy to provide it upon acceptance of the manuscript.

Data Availability and Transparency:

Comment: The data availability statement indicates that full raw data cannot be shared. Given PLOS ONE’s policy, the authors must ensure that the minimal dataset underlying all findings, including numeric values behind summary statistics, is accessible.

Response: Thank you for this comment. We have confirmed with our IRB office that we share the aggregated data and will be happy to provide it upon acceptance of the manuscript.

Comment: The qualitative component would also benefit from transparent reporting of coding materials or at least examples of the codebook and analytic memos, which could be shared as supplementary information without breaching confidentiality.

Response: Thank you for this comment. We have provided the codebook as supplementary information.

---

## [Decision Letter · Decision Letter 2]

1 Jan 2026

Dear Dr. Burke,

Thank you for submitting your manuscript to PLOS ONE. After careful consideration, we feel that it has merit but does not fully meet PLOS ONE’s publication criteria as it currently stands. Therefore, we invite you to submit a revised version of the manuscript that addresses the points raised during the review process.

**ACADEMIC EDITOR:**

Dear Authors,

Thank you very much for your efforts.

I have carefully reviewed the full set of comments provided by Reviewer 1 and 2 across rounds 1, 2, and 3, as well as the authors’ detailed responses and the corresponding revisions made to the manuscript. In addition, I reassessed the comments from Reviewer 1 and the current version of the manuscript in its entirety.

Overall, I consider that the authors have responded adequately to the vast majority of the concerns raised by both reviewers. Many of the major issues identified by Reviewer 1 have been addressed satisfactorily, and the manuscript has improved substantially over successive rounds of revision. Importantly, the authors clearly state that this is an exploratory study. While the manuscript does not employ advanced statistical analyses, the descriptive data are adequately presented, and the study provides a valuable preliminary evidence base upon which future research can build. In my assessment, this exploratory contribution is appropriate for the journal and of potential interest to its readership.

While I agree that some aspects still require correction and clarification, these remaining issues are, in my view, limited in scope and can be addressed through a focused minor revision rather than warranting rejection.

The main points that still need to be comprehensively revised by the authors are confined to the Methods section and the Abstract, specifically:

1. Ensuring that the abstract follows the journal’s required structure for headings (introduction, methods, results, conclusions).

2. Providing a clearer and more explicit description of the methodology, including participant selection procedures, inclusion and exclusion criteria, sampling strategy, sample size justification with appropriate references, and data collection methods.

3. Clarifying the context of the sampled population (prehospital vs. intrahospital emergency settings), as this distinction is important for interpreting disaster management practices.

In light of this, after personal evaluation of all review rounds and revisions, I determine that the remaining concerns can be resolved through a final minor revision, explicitly noting that only three well-defined issues remain to be addressed prior to final consideration for acceptance.

Kind regards,

Javier Fagundo-Rivera

Academic Editor

PLOS One

We look forward to receiving your revised manuscript.

Kind regards,

Javier Fagundo-Rivera, PhD

Academic Editor

PLOS One

Journal Requirements:

**Additional Editor Comments:**

Dear Authors,

Thank you very much for your efforts.

I have carefully reviewed the full set of comments provided by Reviewer 1 and 2 across rounds 1, 2, and 3, as well as the authors’ detailed responses and the corresponding revisions made to the manuscript. In addition, I reassessed the comments from Reviewer 1 and the current version of the manuscript in its entirety.

Overall, I consider that the authors have responded adequately to the vast majority of the concerns raised by both reviewers. Many of the major issues identified by Reviewer 1 have been addressed satisfactorily, and the manuscript has improved substantially over successive rounds of revision. Importantly, the authors clearly state that this is an exploratory study. While the manuscript does not employ advanced statistical analyses, the descriptive data are adequately presented, and the study provides a valuable preliminary evidence base upon which future research can build. In my assessment, this exploratory contribution is appropriate for the journal and of potential interest to its readership.

While I agree that some aspects still require correction and clarification, these remaining issues are, in my view, limited in scope and can be addressed through a focused minor revision rather than warranting rejection.

The main points that still need to be comprehensively revised by the authors are confined to the Methods section and the Abstract, specifically:

1. Ensuring that the abstract follows the journal’s required structure for headings (introduction, methods, results, conclusions).

2. Providing a clearer and more explicit description of the methodology, including participant selection procedures, inclusion and exclusion criteria, sampling strategy, sample size justification with appropriate references, and data collection methods.

3. Clarifying the context of the sampled population (prehospital vs. intrahospital emergency settings), as this distinction is important for interpreting disaster management practices.

In light of this, after personal evaluation of all review rounds and revisions, I determine that the remaining concerns can be resolved through a final minor revision, explicitly noting that only three well-defined issues remain to be addressed prior to final consideration for acceptance.

Kind regards,

Javier Fagundo-Rivera

Academic Editor

PLOS One

Reviewers' comments:

Reviewer's Responses to Questions

**Comments to the Author**

Reviewer #1: (No Response)

Reviewer #2: All comments have been addressed

2. Is the manuscript technically sound, and do the data support the conclusions?

Reviewer #1: No

Reviewer #2: Yes

3. Has the statistical analysis been performed appropriately and rigorously?

Reviewer #1: No

Reviewer #2: Yes

4. Have the authors made all data underlying the findings in their manuscript fully available?

Reviewer #1: Yes

Reviewer #2: No

5. Is the manuscript presented in an intelligible fashion and written in standard English?

Reviewer #1: Yes

Reviewer #2: Yes

Reviewer #1: (No Response)

Reviewer #2: The authors have comprehensively addressed the methodological, presentation, and reporting concerns raised by the reviewers. The manuscript presents an important mixed-methods study, rigorously analyzed in its revised form, and adheres to ethical and data availability standards. The incorporation of a dedicated synthesis section significantly enhances the value of the mixed-methods design.

**Do you want your identity to be public for this peer review?** For information about this choice, including consent withdrawal, please see our Privacy Policy

Reviewer #1: No

Reviewer #2: No

---

## [Author Response · Author response to Decision Letter 3]

20 Jan 2026

Response to Reviewers

We thank the reviewers for the thoughtful and rigorous review. We have carefully addressed each point below and believe these revisions have strengthened the contribution of the manuscript.

The main points that still need to be comprehensively revised by the authors are confined to the Methods section and the Abstract, specifically:

Comment: Ensuring that the abstract follows the journal’s required structure for headings (introduction, methods, results, conclusions).

Response: We have reviewed the journal’s required structure headings and have made the necessary revisions (changed “Purpose” to “Introduction”.)

Comment: Providing a clearer and more explicit description of the methodology, including participant selection procedures, inclusion and exclusion criteria, sampling strategy, sample size justification with appropriate references, and data collection methods.

Response: We have reorganized the Methods section and added explicit subheadings to clarify participant selection and data collection. We have also added a sentence about sample size justification.

Comment: Clarifying the context of the sampled population (prehospital vs. intrahospital emergency settings), as this distinction is important for interpreting disaster management practices.

Response: We have added a sentence that clarifies the context of the sample population.

---

## [Editor Report · Decision Letter 3]

22 Jan 2026

US workforce gaps in emergency management: a mixed-methods approach of demographics, capacity, and community engagement

PONE-D-25-17329R3

Dear Dr. Burke,

We’re pleased to inform you that your manuscript has been judged scientifically suitable for publication and will be formally accepted for publication once it meets all outstanding technical requirements.

Kind regards,

Javier Fagundo-Rivera, PhD

Academic Editor

PLOS One

**Additional Editor Comments:**

Dear Authors,

I am writing to inform you of the outcome of the evaluation of the third revision of your manuscript.

As previously indicated, this third round of revision was assessed directly by me, as the remaining comments and points for improvement concerned three specific and clearly defined issues suitable for resolution through a minor revision.

I have now carefully reviewed your responses and the corresponding modifications to the manuscript. I am pleased to confirm that all comments have been adequately addressed and that the revised version meets the standards required for publication.

I am therefore happy to inform you that your manuscript has been accepted for publication in PLOS ONE. Congratulations on your work.
---

## [Editor Report · Acceptance letter]

PONE-D-25-17329R3

PLOS One

Dear Dr. Burke,

I'm pleased to inform you that your manuscript has been deemed suitable for publication in PLOS One. Congratulations! Your manuscript is now being handed over to our production team.

Kind regards,

on behalf of

Dr. Javier Fagundo-Rivera

Academic Editor

PLOS One